# OpenReview forum: "MedGazeShift : Transferable Multimodal Adversarial Attacks for Diagnostic Misdirection in Vision-Language Models"
_ICLR.cc/2026/Conference — ICLR 2026 Conference Withdrawn Submission_

### Official Review · Reviewer_nP8N · 2025-10-31

**Soundness:** 3
**Presentation:** 3
**Contribution:** 1
**Rating:** 2
**Confidence:** 4

**Summary:**

The paper proposes MedGazeShift, a transferable black-box multimodal attack that misleads medical VLMs by injecting imperceptible perturbations into non-diagnostic regions and shifting model attention away from pathological areas.
It combines background masking (MedSAM), multimodal (image-text) perturbation optimization, and an attention-distract loss.
Experiments on multiple medical datasets and models (GPT-5, Gemini-Pro, MedVLM-R1, etc.) show higher transferability and stealthiness than existing attacks.

**Strengths:**

1. Addresses an important and safety-critical domain (medical AI).
2. Demonstrates consistent empirical improvements across many models and defenses.
3. Introduces domain-aware evaluation metrics (MAS, MTS) reflecting diagnostic plausibility.
4. Offers strong empirical breadth with both open and closed-source VLMs.

**Weaknesses:**

1. Limited novelty. Components (MedSAM masking, multimodal optimization, attention loss) are repurposed from prior work (VLAttack, CWA, etc.).
2. No theoretical or mechanistic insight into why attention shift improves transferability.
3. Mostly an application/benchmark paper, not a fundamentally new attack algorithm.

**Questions:**

Technically solid and experimentally thorough, but conceptually incremental.The mechanism is essentially a log-ratio attention-penalty combined with a CLIP-ensemble optimization setup reused from prior multimodal attack work. The main contribution lies in empirical evidence that medical VLMs are vulnerable, rather than a new methodological advance. Could the authors clarify what is technically novel beyond this recombination of existing components?

---

### Official Review · Reviewer_esxU · 2025-10-31

**Soundness:** 2
**Presentation:** 1
**Contribution:** 2
**Rating:** 2
**Confidence:** 3

**Summary:**

In this paper, the authors proposed MedGazeShift, a transferable multimodal adversarial attack against VLMs (targeting vision-language models) used in the medical diagnostic field. Specifically, it uses MedSAM to identify the foreground with critical pathological information and apply adversarial perturbations to the square areas in the remaining background, such that the  proxy model's attention is forced to move to the background through the optimization of the attention-distract loss. In practice, the authors trained the adversarial samples on an ensemble of 4 ViT-CLIP models and tested their transferability to 6 VLMs, open- and closed-source, demonstrating that MedGazeShift more effectively misdirects models into generating incorrect diagnoses with 3 metrics the authors invented by themselve, i.e. MTR (GPT-judge of whether the generate text is different enough), AvgSim (image similarity to show if the perturbations are too much) and MAS (some weighted combination of AvgSim and attack success rate that seems to somehow relate to MTR). A human evaluation conducted by medical interns rated MedGazeShift highest in adversarial text influence, image quality preservation, and overall attack score. Additionally, ablation studies confirmed the significant contribution of multimodal adversarial representation and attention transfer strategies to the attack’s efficacy and transferability.

**Strengths:**

+ It's nice to see a domain-specific attack.
+ The transferable attack corresponds to realistic usage scenario of medical VLMs.

**Weaknesses:**

+ The writing and the presentation need improvement. For instance, in section 4, the authors introduced an idea of "multimodal seed" when showing an optimization problem for $x',I'$ without ever defining what a seed is, and later in the section $x_\text{seed}$ suddenly appears without explanation of what's its relation with the previous $x'$ and where does $I'$ goes. Likewise, the target embedding $z^*$ was used without any prior explanation until in the Appendix. The authors also failed to explain what the cropping and distortion are in the main body. Basically, the authors shall not write a body that is this non-self-contained and just append an over 30 pages long appendix expecting reviewers and readers to hunt information in it.
+ The dataset and the budget for the attack are questionable.
    + The entire dataset is of size 1k while it contains 7 categories of images and needs to split into train, eval and possible valid as well. The amount of data doesn't seem enough to draw a comprehensive conclusion.
    + Unlike the baselines (e.g. M-Attack) which are designed for general purposed VLM tasks, MedGazeShift is meant for X-Ray, MRI and other medical images that are typically much less diverse. For instance the MRI image in Figure 3 clearly looses too much information after the perturbation with the interior of the brain all burred up, the monochrome images in Figure 12, 20 and 21 become colorful after perturbation which clearly has changed the nature of this medical modality, etc.
+ The specific designs for medical images seems scarce. By replacing MedSAM with a general purpose SAM, I don't see why MedGazeShift cannot be used for general purpose VLM attacks.
+ The idea to break the attention map is not new to VLM attacks. For instance, [1] from 2023 has already explored this method and there certainly are even more classical papers sharing the idea, but when we set aside the manipulation on attention map, the innovative part about MedGazeShift seems little.

1. Yin, Ziyi, et al. "Vlattack: Multimodal adversarial attacks on vision-language tasks via pre-trained models." Advances in Neural Information Processing Systems 36 (2023): 52936-52956.

**Questions:**

Please refer to the weaknesses for my concerns about this paper. The following questions are just for help the authors to understand my concerns.
+ What design in MedGazeShift is unique to medical images? Cannot MedGazeShift compete with the baselines in the regular VLM tasks?
+ Is the budget still acceptable when medical images are much less diverse than the images in typical VLM benchmarking datasets? The perturbation seem to introduce colors to monochrome pictures, distort images where shape matters, remove too much details, etc.?
+ Why is there no examples adversarial images from any baselines?
+ Do you think you can make the main body self-contained given that your appendix looks like a wholly different independent paper of its own?

---

### Official Review · Reviewer_Em9m · 2025-11-04

**Soundness:** 2
**Presentation:** 2
**Contribution:** 2
**Rating:** 2
**Confidence:** 3

**Summary:**

This paper introduces MedGazeShift, a novel and highly transferable black-box adversarial attack specifically designed to compromise medical Vision-Language Models (VLMs). The authors address the critical problem that existing attacks are often unsuitable for medicine because they create visible distortions that a clinician would detect. MedGazeShift's core strategy is to create imperceptible perturbations that cause the VLM to generate a plausible but incorrect diagnosis. The method works by first using a segmentation model to identify the non-diagnostic background regions of a medical image. It then strategically introduces synergistic, multimodal perturbations into this background area. Crucially, the attack uses a novel "Attention-Distract loss" to force the VLM's internal "gaze" or attention mechanism to shift away from the actual pathology and focus on the adversarially perturbed background instead. The authors demonstrate that this attack is highly effective, successfully transferring to six different medical imaging modalities and a range of VLMs, including closed-source models like GPT-5 and Gemini. The paper also introduces new evaluation metrics to jointly measure attack success and image quality preservation, confirming through both automated and human expert evaluation that the attack can successfully misdirect a VLM's diagnosis without a human expert noticing the image manipulation.

**Strengths:**

1. The paper tackles a critical, high-stakes security vulnerability. Its focus on imperceptibility and plausible misdiagnosis makes it more relevant than standard attacks that create obvious visual artifacts or nonsensical outputs.
2. The core idea of hijacking the model's "gaze" by targeting the attention mechanism is sophisticated. The strategy of constraining perturbations only to the non-diagnostic background is a clever way to ensure the attack remains stealthy.
3. The method is shown to be highly effective, consistently outperforming five leading baseline attacks in both automated and human-expert evaluations.

**Weaknesses:**

1. While the attack is "highly transferable," the results in Table 1 show a clear performance gap. The attack is significantly more effective against open-source models (e.g., 0.79 MTR on InternVL) than against closed-source, reasoning-focused models (e.g., 0.48 MTR on Gemini 2.5 Pro and GPT-5).
2. The presentation of this paper is not ideal. It makes this paper hard for me to follow what this paper's motivation and actual contribution are.
3. The primary automated metric for attack success (MTR) relies on an "LLM-as-a-judge" (GPT). This type of measurement is an imperfect proxy for true clinical misdiagnosis and introduces its own potential for bias or error.

**Questions:**

1. Since your attack relies on redirecting attention away from the foreground, have you explored a defense that forces the VLM to base its diagnosis only on the attention scores within the foreground mask?
2. Your attack assumes the background is non-diagnostic. Is this always a safe assumption in medicine?
3. Your ablation study shows that the attention shift and multimodal noise are both critical. How much of the attack's success is from the perturbed text seed versus the image perturbation?

---

### Official Review · Reviewer_TdGK · 2025-11-10

**Soundness:** 2
**Presentation:** 3
**Contribution:** 3
**Rating:** 2
**Confidence:** 4

**Summary:**

This paper proposes MedGazeShift, a transferable black-box multimodal adversarial attack targeting medical VLMs. The method aims to generate imperceptible perturbations in non-diagnostic background regions of medical images, guided by an adversarially crafted text prompt. The attack is optimized using an attention-distraction loss to shift the model's focus away from pathological areas, thereby inducing plausible but incorrect diagnostic reports.

**Strengths:**

The paper focuses on a timely and critical problem: the adversarial vulnerability of VLMs in high-stakes medical applications. Ensuring the robustness of these models is paramount for their safe deployment, and can contribute to understanding potential failure modes.

**Weaknesses:**

1. **Incremental Technical Novelty**: The core technical ideas are adaptations of existing work. Constraining perturbations to the background is a known technique for imperceptibility. The attention loss is explicitly "inspired by" the logarithmic boundary loss from Chen et al. (2020). The multimodal perturbation synthesis is "inspired by" Yin et al. (2023). While combining existing ideas can be a valid contribution, the paper does not adequately differentiate its technical contributions from these prior works, making the novelty appear limited.

2. **Critically Underspecified Methodology**: Key components of the MedGazeShift framework are described at such a high level that the method is irreproducible.

In Section 4.A, the adversarial text generation x_adv = g_phi(I, x) is a black box. There is no description of the model g_phi, how it is trained or prompted, or how the constraint d_text <= eps_text is defined and enforced.
In Section 4.B, the multimodal perturbation synthesis is similarly opaque. The paper states it uses a "joint optimization strategy" and presents L_mm, but the actual iterative process for refining image perturbations and text tokens is not detailed. How are gradients backpropagated through this complex, multi-stage process? What are the architectures of F_alpha and F_beta? Without these details, the method is just a collection of high-level ideas.

3. **Ambiguous Formulation of the Attention Loss**: The attention-distraction mechanism is central to the proposed method, yet its implementation is poorly defined. The paper defines the loss L_attn based on an "attention map h" in Section 4.D. It fails to specify from which layer of the surrogate models this attention map is extracted. Attention maps from different layers of a vision transformer capture vastly different information, from low-level features to high-level semantics. This choice is critical to the method's success and its omission is a significant gap.


4. **Line 195:**
It is unclear how x_{\text{adv}} is generated. The authors should provide a detailed description of the algorithm or specify the exact prompts used to create it.

 **Equation (1):**
The annotation is confusing. The meanings of k, t, i, j are not explained, nor is it clear what y^* represents or where it comes from. The authors should clarify these notations to improve readability and reproducibility.



5. **Figure 3:**
The adversarial examples shown in Figure 3 display visible perturbations on medically critical regions, which contradicts the paper’s claims about preserving these areas. The authors should address this inconsistency and provide further justification or refinement of their defense claim.

6. **Potentially Missing Related Work**

The literature review is insufficient and misses several highly relevant works about VLM safety. Discussing these works would provide crucial context for the paper's contributions.

[a]Effective Black-Box Multi-Faceted Attacks Breach Vision Large Language Model Guardrails https://arxiv.org/abs/2502.05772

[b]SurgVLM: A Large Vision-Language Model and Systematic Evaluation Benchmark for Surgical Intelligence https://arxiv.org/abs/2506.02555

**Questions:**

Refer to the weakness section and:


1. **Attack modality:**
The proposed attack algorithm is described as multimodal. Can a single modality (e.g., text-only) also perform the attack? If not, the authors should explain why multimodality is necessary. Moreover, attacks conducted on the image modality should be justified as essential to the overall framework.

2. **Background-constrained perturbation:**
What happens if most or all regions of a medical image are critical areas (e.g., pathology images)? In such cases, the permissible background area for applying perturbations would be extremely limited. The authors should discuss the feasibility and potential performance degradation in such scenarios.

---

### Note · Authors · 2026-01-11

I have read and agree with the venue's withdrawal policy on behalf of myself and my co-authors.